# Novel Insights into Changes in Gene Expression within the Hypothalamus in Two Asthma Mouse Models: A Transcriptomic Lung–Brain Axis Study

**DOI:** 10.3390/ijms25137391

**Published:** 2024-07-05

**Authors:** Eslam M. Bastawy, Izel M. Eraslan, Lara Voglsanger, Cenk Suphioglu, Adam J. Walker, Olivia M. Dean, Justin L. Read, Mark Ziemann, Craig M. Smith

**Affiliations:** 1Faculty of Health, School of Medicine, Institute for Mental and Physical Health and Clinical Translation (IMPACT), Deakin University, Geelong 3216, Australia; e.ahmed@deakin.edu.au (E.M.B.); ieraslan@deakin.edu.au (I.M.E.); lvoglsan@deakin.edu.au (L.V.); a.walker@deakin.edu.au (A.J.W.); justin.read@monash.edu (J.L.R.); craig.smith@deakin.edu.au (C.M.S.); 2Faculty of Science, Engineering and Built Environment, School of Life and Environmental Sciences, Deakin University, Geelong 3216, Australia; cenk.suphioglu@deakin.edu.au (C.S.); mark.ziemann@burnet.edu.au (M.Z.); 3Florey Institute of Neuroscience and Mental Health, The University of Melbourne, Parkville, Melbourne 3052, Australia; 4Burnet Institute, Melbourne 3004, Australia

**Keywords:** asthma, mental health, lung–brain axis, hypothalamus, RNA-seq

## Abstract

Patients with asthma experience elevated rates of mental illness. However, the molecular links underlying such lung–brain crosstalk remain ambiguous. Hypothalamic dysfunction is observed in many psychiatric disorders, particularly those with an inflammatory component due to many hypothalamic regions being unprotected by the blood–brain barrier. To gain a better insight into such neuropsychiatric sequelae, this study investigated gene expression differences in the hypothalamus following lung inflammation (asthma) induction in mice, using RNA transcriptome profiling. BALB/c mice were challenged with either bacterial lipopolysaccharide (LPS, *E. coli*) or ovalbumin (OVA) allergens or saline control (n = 7 per group), and lung inflammation was confirmed via histological examination of postmortem lung tissue. The majority of the hypothalamus was micro-dissected, and total RNA was extracted for sequencing. Differential expression analysis identified 31 statistically significant single genes (false discovery rate FDR5%) altered in expression following LPS exposure compared to controls; however, none were significantly changed following OVA treatment, suggesting a milder hypothalamic response. When gene sets were examined, 48 were upregulated and 8 were downregulated in both asthma groups relative to controls. REACTOME enrichment analysis suggests these gene sets are involved in signal transduction metabolism, immune response and neuroplasticity. Interestingly, we identified five altered gene sets directly associated with neurotransmitter signaling. Intriguingly, many of these altered gene sets can influence mental health and or/neuroinflammation in humans. These findings help characterize the links between asthma-induced lung inflammation and the brain and may assist in identifying relevant pathways and therapeutic targets for future intervention.

## 1. Introduction

Several recent systematic reviews have demonstrated that diverse mental illnesses such as bipolar, major depressive disorder (MDD) and anxiety disorders are significantly associated with high serum levels of inflammatory cytokines [1,2,3,4,5] and systemic inflammation [6]. Although the relationship is bidirectional, peripheral inflammation can be a strong driver of poor mental health and neurological disorders [7,8,9,10,11,12,13]. This study investigates this further by characterizing the transcriptomic changes within the hypothalamus in two mouse models of asthma.

Throughout the last twenty years, there has been growing recognition of the lungs as a crucial peripheral site where inflammation can impact mental health [14,15,16]. However, inflammatory lung–brain interactions still receive little attention when compared with the inflammatory gut–brain axis. Indeed, recent decades have witnessed an exponential growth of evidence detailing the interplay between the gut and the brain [17] and its influence on mood and mental health [18,19]. In contrast, while lung inflammation may be similarly important, it remains relatively less well explored.

Asthma is a worldwide noncommunicable disease [20] and can significantly impair an individual’s overall quality of life [21,22]. Asthma can be broadly categorized into two primary types: allergic and non-allergic asthma. Allergic asthma, the most common form, accounts for more than 60% of asthma cases [23] and is triggered by exposure to different allergens including pollen, dust mites and tobacco smoke. The pathophysiology of allergic asthma involves a T-helper cell 2-immune response causing airway hyperresponsiveness and eventually chronic inflammation and narrowing of the pulmonary airways. As a result, asthmatic individuals may experience frequent fatigue episodes, shortness of breath, coughing, wheezing and chest tightness [24,25].

As well as acting locally within the lungs, cytokines and other pro-inflammatory hormones can enter the bloodstream and travel to the brain [13]. The hypothalamus is considered the quintessential brain region that is adversely affected by peripheral inflammation [26], as parts of the hypothalamus, including the median eminence and arcuate nucleus, are not protected by the blood–brain barrier [27]. The hypothalamus, a key regulator of systemic homeostasis [28], plays a substantial role in mental health and neurological diseases [29,30]. For example, recent studies show that patients with depressive disorders have a larger hypothalamus [31]. Peripheral inflammation can cause neuroendocrine, neuronal and metabolic perturbations in this brain structure, which has recently shown that hypothalamic neuroinflammation is linked to prior stress exposure and subsequent worsening of mental health [32,33].

To gain insights into the neurological changes that occur in response to lung inflammation, we established two allergic asthma models and compared their hypothalamic transcriptome response with control mice. Typically, airway inflammation characteristic of allergic asthma is provoked by sensitizing and challenging mice using a foreign antigen. In the present study, we employed two kinds of allergens in two separate groups of mice: chicken egg ovalbumin (OVA) [34,35] and the environmentally ubiquitous lipopolysaccharide (LPS) which is a major component of the cell membrane of Gram-negative bacteria [36]. The neurological response within the hypothalamus was then characterized using whole-transcriptome sequencing to determine the differences in gene expression caused by asthma.

## 2. Results

### 2.1. Confirmation of Lung Inflammation in OVA- and LPS-Treated Mice

Two typical pathological signs of asthma were detected in both asthmatic OVA- and LPS-treated groups, relative to controls. Firstly, H&E staining revealed increased inflammatory cell infiltration around blood vessels (perivascular), bronchi (peribronchiolar) and nearly all other components of lung tissue (Appendix A). When these changes were qualitatively assessed, both LPS and OVA groups recorded equivalent inflammation scores that were significantly greater than controls (Appendix A). Secondly, Masson’s trichrome staining revealed signs of increased fibrosis and sub-epithelial collagen deposition in the perivascular and the peribronchiolar regions of asthmatic lungs, compared to saline-challenged controls (Appendix A).

### 2.2. Validation of the Microdissection

After collecting each circular biopsy sample, sections were imaged in order to confirm accurate anatomical collection. As an additional measure to confirm the reliability of our collected areas, we identified genes to use as regional markers that are expressed either within our area of collection or just outside our targeted area of collection, i.e., within the hypothalamus, lateral boundaries of our collection (amygdala) and dorsal boundaries (the thalamus) (Appendix A). The neuroanatomical distribution of cells expressing these genes was determined using in situ hybridization (ISH) data from the Allen Brain Atlas and the Gene help of NCBI’s databases [37].

### 2.3. Bioinformatics Analysis

#### 2.3.1. Single Genes Differentially Expressed in Asthma Groups vs. Control

To better understand the effect of asthma on whole-genome hypothalamic gene expression, RNA-seq was performed on the hypothalamus isolated from serial coronal sections of mouse brains after challenging BALB/c mice with either LPS or OVA. Following the alignment and quantification of gene read expression, differential expression analysis of single genes and gene sets was conducted. There were 16,100 and 17,144 genes detected in the LPS- and OVA-treated groups, respectively. Multidimensional scaling analysis reveals that asthma challenge had only a modest effect; relatively few genes showed significant differences between treatment groups. As a result, highly sensitive analysis involving gene set enrichment, the Mitch method, was applied to identify subtle differences in the hypothalamus response to asthma challenge (Appendix A). Then, a two-dimensional rank–rank plot was used to compare gene expression changes in OVA and LPS asthma groups relative to controls (Appendix A). Most genes were located in either the bottom-left or top-right quadrants, indicating that these stimuli cause similar changes to global gene expression patterns. Volcano plot analysis of the LPS-challenged group demonstrated that 31 single genes met the threshold for statistical significance (FDR < 5%) and were differentially expressed versus control (15 upregulated and 16 downregulated; Appendix A). In contrast, when OVA-treated mice were compared to controls, no single genes met the threshold for statistical significance (FDR < 5%) (Appendix A). This threshold was more stringent in this data set, as it was calculated separately and depended on multiple different factors such as sequencing depth and purity and differences in the number of genes detected in each group.

Next, we conducted clustering heatmap analysis to visually depict data from each mouse for the top 50 genes, as ranked by statistical significance/*p* value. LPS versus control data displayed relatively consistent visual trends, whereby mostly all mice within the LPS group were similarly increased (red) or decreased (blue) for each gene, while control mice displayed mostly consistent opposite directionality for each gene (Appendix A). In contrast, higher within-group variability was visually obvious in OVA versus control data (Appendix A), which likely explains the higher *p* values recorded for this asthma treatment and the absence of statistically significant changes in single gene expression versus control. Furthermore, no single gene appeared in the top 50 list for both asthma treatments. Therefore, we did not perform any further analysis of single genes and instead focused on gene set analysis.

#### 2.3.2. Gene Sets Differentially Expressed in Asthma Groups vs. Control

Gene set enrichment analysis was performed (R statistical software package “Mitch (1.8.0)”), detecting 1179 REACTOME gene sets in LPS-challenged mice, of which 80 gene sets were significantly (<FDR5%) upregulated, and 25 were downregulated compared with control mice (Figure 1a). In contrast to the single gene analysis, OVA treatment caused a greater number of gene sets to be differentially expressed compared to controls. Of the 1178 REACTOME gene sets detected, 112 were significantly upregulated, and 51 were downregulated (FDR<5%, Figure 1b). Many of these upregulated gene sets recorded extremely low *p*-values, causing visual vertical compression of the remaining data in the resulting volcano plot. Interestingly, 48 upregulated and 8 downregulated gene sets commonly overlapped between both asthma treatments (Figure 1c). A detailed description of the significant gene sets is available in Appendix A. Next, the gene sets with the greatest magnitude of change in expression (top 10 increased expression and top 10 decreased expression) were identified following LPS treatment (Appendix A) and OVA treatment (Appendix A), relative to control.

As an additional way to determine the gene sets that exhibit joint enrichment across the two asthma models, we performed two-dimensional Mitch analysis compared to the control group. When depicted in a scatter plot (Appendix A) as expected, it was visually obvious that most gene sets were within either the top-right quadrant (corresponding to increased expression in both asthma groups relative to control) or the bottom-left quadrant (corresponding to decreased expression in both asthma groups relative to control), although some red points (gene sets) were clustered in the discordant quadrants. Of the 1175 gene sets included, 205 were statistically significant (FDR < 0.05). Next, the top 50 gene sets ranked by magnitude of change (i.e., s-distance, or distance from scatterplot axis origin, also referred to as the enrichment score), regardless of the statistical significance, were identified and depicted as a heatmap (Figure 2). Detailed information on the enrichment scores and *p* values for the top significant gene sets in both LPS and OVA asthma treatment is available in Appendix A.

In order to gain a comprehensive understanding of the functions and impacts of the most significant gene sets, our approach involved multiple steps for analysis. Firstly, we used the RECTOME database to unveil the functional upstream hierarchy that governs these gene sets, subsequently categorizing them based on their shared traits. For instance, the prominent dysregulated gene sets exhibited functional links to diverse processes such as metabolic regulation, cellular response to stimuli, gene expression modulation, cell cycle progression and homeostasis maintenance. Interestingly, among the top 50 dysregulated gene sets, 5 were categorized under neuronal system signaling and were downregulated in both LPS- and OVA-challenged groups. These include *highly calcium permeable postsynaptic nicotinic acetylcholine receptors, acetylcholine binding and downstream events, serotonin neurotransmitter release cycle, CREB1 phosphorylation through the activation of adenylate cyclase* and *voltage gated potassium channels.* This highlights a specific brain immune response to asthma, indicating that asthma may not only be a respiratory condition but also one that involves complex neuro-immune interactions, as shown in Figure 3.

Furthermore, when gene sets were prioritized by discordance to reveal those with opposing expression levels between the LPS- and OVA-challenged groups (Appendix A), four gene sets directly involved in the immune system were identified. These included gene sets closely associated with adaptive immune responses, such as *immunoregulatory interactions between lymphoid and non-lymphoid cells*, and the *endosomal vacuolar gene set*, as well as those involved in both adaptive and innate immune responses, such as *interferon gamma signaling* and *interferon alpha/beta signaling* gene sets. Interestingly, these gene sets were upregulated in response to OVA challenge and downregulated in the LPS-challenged group. The contour plots illustrate this by showing a concentration of dark red in the upper right and left quadrants, indicating several genes significantly upregulated in the OVA challenge compared to the LPS challenge. This is consistent with a stronger or more pronounced adaptive immune response to OVA. This variation could be attributed to the differing immune responses to LPS and OVA (Figure 4).

Secondly, we conducted an extensive review of the existing literature to delve into the downstream effects stemming from the top concordant significant gene sets. Intriguingly, our investigation revealed that certain pathways within these gene sets had direct correlations with mental health. Notably, some of these pathways demonstrated overlaps between neuroinflammation and mental health, suggesting intricate interplays that could have far-reaching implications (Figure 5).

## 3. Discussion

An ever-growing range of studies has highlighted the significant psychological impact of the inflammatory changes within the lung conditions on mental health, as recently seen with the severe acute respiratory syndrome coronavirus 2 (SARS-CoV-2) COVID-19 pandemic [38,39] and asthma [40]. However, the mechanisms through which lung inflammation affects the brain and influences mental health are not yet fully characterized. This study aimed to better understand the molecular crosstalk between the lung and the hypothalamus as it is preferentially affected by peripheral inflammation. This study serves as a new reference for the transcriptomic changes in the hypothalamus in response to these two commonly used asthma models. Our findings provide insights into the neurological mechanisms underlying changes in the hypothalamus caused by asthma. Moreover, these findings also assist in determining lung–brain pathways that are critical to future drug discovery and improving mental health.

A notable finding is the significant downregulation of various gene sets involved in hypothalamic neuronal system signaling, observed in response to both LPS and OVA challenges. Many of these gene sets have established links to mental health disorders. Specifically, our results indicated decreased expression of the *CREB1 phosphorylation through the activation of adenylate cyclase*, a pathway primarily involved in activating the CREB-regulated transcription coactivator 1 (CRTC1). CRTC1 signaling has diverse functions in the brain, including roles in synaptic plasticity and mood regulation, and it is also associated with several mood disorders [41,42,43,44]. Furthermore, our results are consistent with previous studies on depressive preclinical mouse models, which demonstrated that life stressors dampen CRTC1 expression in the prefrontal cortex and hippocampus. This leads to disruptions in functional connectivity and neural plasticity, both of which are associated with the pathogenesis of depression [45]. Additional evidence from a prenatal stress model in rats shows that the offspring exhibited depressive-like behaviors associated with downregulated CRTC1 signaling [46]. Furthermore, CRTC1 has been identified as a key modulator of the antidepressant response, evidenced by its increased expression following antidepressant intervention [47], and plays a central role in regulating food intake through its action on orexigenic hypothalamic genes. Its deficiency affects circadian rhythms, demonstrated by increased diurnal locomotor activity in male CRTC1^−/−^ mice [45]. Moreover, CRTC1 signaling is considered to act as a potent regulator of normal brain energy metabolism, with studies showing that its deletion results in impaired hippocampal metabolism, a fundamental cause of depressive-like behaviors [48]. In addition, multiple lines of evidence highlight the critical role of CRTC1 in linking the comorbid neurobiological mechanisms of depression and obesity [49]. Additionally, previous human studies have noted reduced CRTC1 expression in the postmortem brains of depressed patients [50,51] and suicide victims [52]. All these pieces of evidence suggest that asthma-induced CRTC1 downregulation may affect hypothalamic metabolism and the circadian rhythm and disrupt neural plasticity, thereby predisposing individuals to mental health disorders.

Transitioning to the cholinergic system, it is recognized for its essential role in the pathophysiology of mood disorders [53]. Our study showed a downregulation of certain cholinergic gene sets including the *acetylcholine binding and downstream events*, suggesting a potential role in mental health. This is substantiated by evidence that cholinergic dysregulation can result in anxiety and depressive-related symptoms [54]. Another noteworthy gene set, the *highly calcium permeable postsynaptic nicotinic acetylcholine receptors*, is implicated in long-term potentiation, and its dysregulation may contribute to synaptic dysfunction causing depressive disorders [55,56,57]. Furthermore, investigation into the serotonergic neurotransmitter system identified the decreased transcription of the *serotonin neurotransmitter release cycle* gene set within the hypothalamus, which is hypothesized to be attributed to the decreased serotonergic input to the corticotrophin-releasing hormone (CRH)-containing cells in the paraventricular nucleus, suggesting the decreased number of serotonin presynaptic vesicles and in turn synaptic dysregulation. Pertinent to this, presynaptic vesicles were markedly decreased in an asthma animal model when assessed using transmission electron microscopy [58]. Furthermore, this study also showed that the *voltage gated K^+^ channel* gene set was significantly dysregulated in both asthma treatments. Voltage gated K^+^ channels play a vital role in neural plasticity, action potential modulation and neurotransmitter release homeostasis, and their alteration is associated to several neuropsychiatric disorders including depression [59], bipolar disorder, schizophrenia, autism spectrum disorder (ASD) and autoimmune psychiatric disorders [60].

In addition, of the top 48 significant gene sets that were identified in response to both asthma treatments, most of them are involved in several key processes that are relevant to mental health. Firstly, both asthma groups showed reduced expression of the *class c metabotropic glutamate receptor* gene set which is directly involved in the regulation of glutamatergic neurotransmission. This finding is consistent with human brain transcriptome studies that also observed reduced expression of this same gene set in depressed and suicide participants compared to controls, within samples from the dorsolateral prefrontal cortex [61] and other cortical and subcortical regions [62]. Moreover, a postmortem study using PET scans and Western blotting linked a reduced density of metabotropic glutamate receptors to major depressive disorder [63]. Consistent with these studies, a recent systematic review of proton magnetic resonance spectroscopy studies [64] confirmed that a dysregulation of glutamatergic neurotransmission is associated with depression.

Secondly, our transcriptomic data showed the overexpression of a pathway related to protein metabolism, the *SRP-dependant co-translational protein translocation,* referred to as ER translocation genes. Interestingly, a recent meta-analysis examining data from Alzheimer’s patients and mouse models across various brain areas demonstrated that the overexpression of ER translocation genes is linked to brain hypometabolism and disruption of the protein life cycle, correlating with cognitive impairment [65].

Thirdly, the p75 neurotrophin receptor (p75^NTR^) is characterized as a low-affinity receptor capable of binding to various neurotrophic factors, including BDNF and nerve growth factor. Its main function is the modulation of synaptic plasticity and transmission, maintenance, survival and death of neurons [66]. Our results showed the downregulation of the p75^NTR^-related genes, which is postulated to be concomitant with mood disorders. Such a claim was supported by a previous study that showed p75^NTR^-related allele (L205) was significantly less expressed in depressed and suicide victims, suggesting its protective effect against depression pathogenesis [67]; such p75^NTR^ disturbance and mood disorder associations were reviewed by [68]. Moreover, in the mice hippocampus, the functional role loss of p75^NTR^ as shown in p75^NTR−/−^ animals presented with behavioral deficit baseline anxiety and difficulties in recovering from stress [69,70]. Additionally, an in vitro study using cultured hippocampal neurons has demonstrated that p75 neurotrophin receptor is critical for TrkB functional signaling in regulating neuronal function and survival [71]. Other lines of research show that p75^ntr^ null mice experienced a significant reduction in synaptic vesicles in motor axon terminals [72]. Such neural plasticity disturbance has been proposed to account for the molecular mechanisms underlying psychiatric disorders [73]. Furthermore, a recent genome-wide pathway analysis, building on previous work published over 20 years, has highlighted the involvement of p75^NTR^ signaling in acute brain injuries and plasticity, as well as in neuronal cell damage and neurodegeneration [74].

In addition, downregulation of the *activator protein 1 (AP-1) family* gene set was observed in both OVA and LPS models. AP-1, a transcription factor intricately linked to the modulation of gene expression in the nervous system, plays a crucial role in neural plasticity [75]. As a result, low levels of the AP-1 transcription factor are associated with dysfunctional plasticity, thereby contributing to neuropsychiatric disorders [76]. Furthermore, an in vivo study conducted to investigate the mode of action of the widely used neuropsychiatric treatment, the selective serotonin reuptake inhibitor (SSRI) drug fluoxetine, has shown that the AP-1 transcription factor is essential for the antidepressant response and also regulates key neural plasticity genes associated with depression [77]. On the other hand, lower AP-1-family-related gene expression appears to contradict other findings of increased AP-1 gene expression during neuroinflammation [78].

In the same vein, the *Netrin-1/DCC guidance* gene set, otherwise known as commissural axon pathfinding, contributes significantly to synapse formation and plasticity via two main components, Netrin-1 (ligand) and its receptor DCC (deleted in colon cancer). Netrin-1 directs the growth of DCC-expressing axons, influencing axonal navigation and arborization [79]. The recent review of stress mice models, genome-wide association (GWAS) and postmortem human brain studies highlighted the importance of Netrin-1 and its DCC receptor in the prefrontal cortex as an important molecular pathway involved in MDD pathophysiology. In support of this link, our results showed dysregulation of the axon pathfinding gene pathway. We hypothesize that asthma causes alterations in the hypothalamic architecture and function with subsequent depressive implications and other psychiatric disorders [80,81].

On top of that, *nitric oxide stimulates guanylate cyclase*, *cyclic GMP effects*, and *reduction of cytosolic Ca (NO/sGC/cGMP)* gene sets have been considered in memory formation. Usually, nitric oxide stimulates the soluble guanylate cyclase (sGC) and cyclic guanosine monophosphate (cGMP), which in turn, stimulate the long-term potentiation process (LTP), thereby enhancing memory formation processes in the hippocampus [82]. Interestingly, previous studies showed decreased NOS in the paraventricular nucleus of the hypothalamus of depressed patients [83,84]. Moreover, it is worth mentioning that the NO-sGC-CGMP pathway is the main driving force for cholinergic, glutaminergic and dopaminergic signaling [85]. Recently, some researchers explained the association between the neuroplasticity controlled by NO and its antidepressant effect [86]. Additionally, the NO signaling pathway was recently shown to regulate anxiety-related behavior in mice treated with lithium [87]. In line with this, these gene sets including the *NO-sGC*, *reduction of cytosolic Ca* and *cGMP effects* gene sets were also associated with mental health as previously reported in schizophrenia treatment [88], anti-depressive effects [86], pharmacological intervention for enhancing learning and memory formation and performance [85,89,90,91] and neuroprotective effects [92].

In addition to changes in gene-set expression that may underlay mental health problems, we also detected altered expression of several gene sets that have been linked to neuroinflammation. Firstly, both asthmatic groups showed a downregulation of homeostatic gene sets, including *nitric oxide stimulated guanylate cyclase* and *CGMP*, which play a crucial role in the signaling pathways associated with neuroinflammation, including the release of inflammatory mediators, cell migration and the activation of immune cells, as highlighted using in vitro and in vivo models [92]. Moreover, the *reduction of cytosolic calcium (Ca^2+^)* gene set has been previously linked to intracellular calcium dysregulation and neuroinflammation [93]. In addition to neuroinflammation, all these gene sets have also been involved in other functions such as neurotransmission [94].

Secondly, LPS and OVA are known to promote cellular oxidative stress [95,96,97,98]. Interestingly, our RNA-seq results indicated a presence of cellular stress state and subsequent endoplasmic reticulum (ER) stress, as shown by the upregulation of several stress gene sets. Importantly, this included a conserved RNA surveillance gene set, the *NMD*, [99] and the *α-subunit of eukaryotic translation initiation factor 2 “eIF2α”*. Their activation is believed to diminish the abnormal translation rate of accumulated unfolded proteins and hence control chaotic cellular stress [100]. Pertinent to this, previous reports indicated similar activation of the unfolded protein response to ER stress using an in vitro periodontitis model [101] and both in vivo mouse and in vitro lung inflammation models [102]. A similar pattern of ER stress was observed in the lungs of mice subjected to a similar OVA challenge as employed here [103]. The dysregulation of the *NMD* gene set suggests a potential impairment of the RNA quality control mechanisms in the hypothalamus, which has been previously demonstrated to result in neurophysiological and behavioral changes in mice forebrain using a KO mice model [104]. This dysregulation may also be relevant to various neurodegenerative and neurodevelopmental disorders [105]. In line with the cellular stress induced by LPS and in concordance with our transcriptome data, recent findings have indicated the upregulation of a distinctive gene set, the *SRP-dependant co-translational protein translocation*, which is also associated with ER cellular stress [106]. The dysregulation in the endoplasmic reticulum has been extensively reported to be associated with a variety of neuropsychiatric disorders [107].

Thirdly, our results showed a significant downregulation of additional gene sets linked to DNA integrity and repair mechanisms, for instance, the *homology-directed repair (HDR)* pathway, which has recently been altered in response to both LPS and OVA asthma treatments [108], and the *apoptosis-induced DNA fragmentation* gene set, which is directly associated with genomic instability [109,110]. Several studies have reported that different neuropsychiatric conditions show significantly high levels of DNA damage [111]. For instance, increased DNA fragmentation has been reported in the anterior cingulate cortex [112], in addition to elevated levels of single/double strand breaks within the hypothalamus and other brain regions of individuals with bipolar disorder [113].

Strikingly, our functional enrichment analysis displayed that gene sets were commonly involved in both mental health and neuroinflammation. For instance, gene sets of mitochondrial metabolism were significantly overexpressed including the *formation of ATP chemiosmotic coupling*, *complex I biogenesis*, *respiratory electron transport* and *mitochondrial translation* gene sets. The demands of continuous ATP energy and any metabolic dysregulation are considered a major pathophysiological mechanism in major depressive disorder [114,115,116]. Moreover, recent findings explained the shared molecular links between depression and certain neurological diseases, such as Alzheimer’s, involving impaired brain energetics [117,118]. In support of this notion, some researchers suggested that dysregulated expression of ATP biosynthesis genes was among individuals who experienced major depression and suicide, specifically in the ventral prefrontal cortex [61,119]. Interestingly, mitochondrial dysfunction of glial cells has been linked to the excessive production of the mitochondrial reactive oxygen species (mtROS) and ATP release, leading to the activation of the NLRP3 inflammasome and the subsequent activation of caspase-1 inflammatory response causing neuroinflammation [120].

In addition, our Mitch enrichment analysis showed that biological processes related to protein metabolism were significantly upregulated including translation gene sets *eukaryotic translation initiation-cap dependant (eIFS)*, *eukaryotic translation initiation, eukaryotic translation elongation*, *SRP-dependant co-translational protein targeting and translation pathways* and *protein folding* gene sets including *formation of tubulin folding by CCT/TRIC* and *post-chaperonin tubulin folding pathways*. These results suggest a chaotic state of translational dysregulation and protein misfolding. Interestingly, recent reviews briefly discussed the role of protein misfolding in neurons and synaptic connection dysfunction and subsequent brain damage [121], and mRNA-impaired translation is considered a common hallmark for the progression and development of psychiatric diseases such as depression [122,123,124], in addition to the involvement of specific translational gene pathways as *eukaryotic translation initiation* in depression and neuroinflammation [125,126].

Furthermore, our results indicate glycosaminoglycan (GAC) dysregulation, specifically, *keratan sulphate degradation gene set*. GACs are considered major constituents of the brain extracellular matrix and are not only responsible for brain tissue healing and homeostasis but also involved in synaptic plasticity, inflammation and psychiatric illness [127]. Additionally, similar research results were associated with neurodegenerative hippocampal alterations causing Alzheimer’s disease [128].

The difference between the responses to LPS and OVA challenges lies in the nature of the antigens, the type of immune responses they elicit and the pathways involved in regulating these responses. LPS primarily acts as a potent trigger for the rapid and non-specific innate immune response [129]. In contrast, OVA, a T-cell-dependent protein antigen, commonly engages more specific adaptive immunity pathways associated with antigen processing and presentation, as well as T-cell activation and differentiation, particularly Th2-type responses [130,131]. Our Mitch discordant analysis revealed four gene sets intimately associated with the adaptive immune system. These gene sets were notably upregulated in response to the OVA challenge, consistent with the expected adaptive immune response to a protein antigen. Among these, the gene sets related to *immunoregulatory interactions between lymphoid and non-lymphoid cells* and those involved in *endosomal/vacuolar processes* are particularly crucial in orchestrating antigen-specific immune responses [132]. This upregulation highlights the specific engagement and modulation of pathways crucial for the adaptive immune system in response to OVA, aligning with the known biology of how OVA challenges stimulate a complex and specific immune response.

It is interesting that, in contrast to the gene sets where a large number of gene sets were significantly altered in both asthma treatments compared to the control, no single genes were significantly altered in both. This may highlight specific differences in the mechanisms through which OVA treatment versus LPS treatment act on the brain. However, this finding also possibly underscores the power of the mitch gene set approach, where pathways of interest can be identified even in the absence of similarly significant changes in single gene expression. Although each asthma group revealed a different list of significantly altered single genes, this does not automatically mean that those single genes aren’t important for further study (although doing so is out of the scope of the present study). An important first step in the further investigation of these single genes would be to validate their differential expression using qPCR (mRNA) and/or Western blotting (protein levels) in the case of highly expressed genes. Given the often restricted regional/sub-cellular expression of these genes, identifying the specific hypothalamic nuclei that exhibit changes in response to asthma is also crucial. In this regard, RNAscope can play a pivotal role in confirming asthma-induced changes in hypothalamic pathways (such as hypothalamic–pituitary hormonal axes) relevant to mental health. In the current study, we investigated the hypothalamic response in asthmatic male mice exclusively, as most previous similar studies have used males due to their lower variability compared to females. Although existing evidence suggests that neuroinflammatory processes are highly similar in both sexes, the rates of depression are twice as prevalent in females. Additionally, asthma is more common in females, potentially leading to more pronounced hypothalamic responses. Therefore, extending the current studies to perform similar analysis on female mice to identify specific processes is important.

Another crucial future direction is to evaluate the translational relevance of the current mouse studies. Replicating this study in humans poses significant challenges due to the difficulty of obtaining human brain samples. Postmortem samples are often collected after extended periods, sometimes several days, by which point mRNA degradation is likely. Instead, genetic studies using cheek cell samples (for example), where genetic mutations are also present in neurons, offer a more feasible approach. Like many neurological diseases, this approach can allow investigations of whether variations in genetic predisposition confer some individuals to be more vulnerable to asthma-induced mental health issues while others remain resilient. Better understanding these genetic variations could provide crucial insights into individual susceptibility and resilience, ultimately opening new avenues to develop pharmacological drugs that help prevent some of the mechanistic changes observed in the present study.

Finally, while the findings of this study provide valuable insights into the brain’s response to asthma, we note several limitations of the study. Firstly, we used bulk RNA-seq methodology so we do not know conclusively whether the changes observed are due to cell-population differences or differences in the expression levels of these cells, making it challenging to pinpoint specific cellular contributions to the observed alterations. As such, more refined techniques such as single-cell RNA sequencing would be advantageous for dissecting the intricate cellular heterogeneity underlying the reported findings. Moreover, for inflammatory stimuli, it is crucial to consider the potential activation of immune cells, particularly microglia and astrocytes, which could potentially modulate the local environment within the brain, infiltrate several brain regions and contribute to the observed gene expression patterns. While the current study provides a comprehensive snapshot of the transcriptomic landscape, further investigation targeting mice behavior may help to validate our molecular findings underlying the lung–brain axis.

We interpreted our observed changes in the brain as being primarily due to the local inflammation (asthmatic changes) within the lungs. However, it is possible that some of these changes in the brain are caused by the prior injection of LPS or OVA during the sensitization phase itself, as numerous studies have shown that IP injection of LPS can alter the brain [95]. To determine the relative contribution of LPS/OVA injection during the sensitization period versus LPS/OVA inhalation during the challenge phase, future studies could include additional control groups that only receive LPS/OVA sensitization injections followed by subsequent saline inhalation. It would also be of interest to visualize any subtle neuroinflammatory changes between the sensitization, challenge and combined phases using markers of microglia and astrocytes, such as ionized calcium-binding adaptor molecule 1 (Iba1) and glial fibrillary acidic protein (GFAP). Therefore, conducting an expanded study that includes these additional control, sensitization and challenge groups is an important future direction. It is, however, important to note that both antigen sensitization and subsequent challenge phases are required to induce lung inflammation/asthma in mice and that just one phase alone is not sufficient [35,133]. For example, Shin et al. (2022) [134] showed that antigen sensitization is required for Th2 cell activation and recruitment of eosinophils to the lungs, by measuring the BAL fluid of mice sensitized with adjuvant (alum + OVA), non-adjuvant (PBS + OVA) and negative control (no adjuvant or OVA). In addition, the non-adjuvant group (PBS + OVA) could elicit an immune response in the lungs after vaporized allergen exposure independent of the adjuvant but at a significantly reduced level. This endorses the findings of Kool et al. (2008) [135] that soluble OVA is taken up by dendritic cells via afferent lymphatics, whereas after IP injection of alum adjuvant + OVA, the OVA is taken up, processed and presented by inflammatory monocytes, thus becoming inflammatory dendritic cells that induce a persistent Th2 adaptive response. Therefore, any effect caused by the subsequent re-exposure via nebulized OVA, which is breathed into the lungs, would be independent of those caused by the adjuvant.

In addition to the limitations of only investigating male mice and the absence of qPCR validation of single gene changes, another important limiting factor is determining the applicability of our findings to other peripheral inflammatory disorders. To address this, parallel groups of mice could be subjected to a gut inflammatory protocol by feeding them OVA or LPS. Another limitation is that we interpreted our changes in the brain as being primarily due to and downstream of the inflammation in the lungs causing neuroinflammation. However, it is also known that these asthma treatments cause reduced lung function and a range of behavioral changes. Therefore, our current study doesn’t allow us to determine whether the brain changes are due to the effect of blood-borne cytokines or sensory/autonomic lung–brain inputs or whether they are indirect, long-term downstream changes resulting from reduced lung capacity and the associated behavioral changes. This question is also present within human epidemiological studies, where it is not known whether asthma worsens mental health due to neuroinflammatory mechanisms or because asthma discourages exercise and other behavioral and social contact, which are known to benefit mental health.

## 4. Materials and Methods

### 4.1. Animals

Twenty-one male BALB/c mice (n = 21) were purchased from the Animal Resource Centre (Perth, Australia) at 4 weeks of age. This strain is primarily used in the literature for generating allergic asthma models, as they respond well to LPS and OVA sensitization due to their strong immunological response characterized by a bias toward T helper cell 2 (Th2) [136,137]. The mice were group-housed (4 per cage), acclimatized for 1 month in a specific germ-free (SPF) environment and maintained at a temperature of 24 ± 1 °C and 12/12 h light–dark cycle, with water and standard mouse chow ad libitum. All animal procedures were approved by the Deakin University Animal Ethics Committee (project ID: G16-2020) and were in line with the National Health and Medical Research Council (NHMRC, Melbourne, Australia) guidelines for the human handling and care of animals.

### 4.2. Experimental Timeline

Experimental design was constructed from previously reported LPS and OVA asthma models [138,139]. Mice were randomly separated into three experimental groups (n = 7/group): one control group and two asthma groups. The LPS asthma group was sensitized to LPS by IP injection (100 µg/kg, *E. coli*, serotype 026: B6), once per week for 3 consecutive weeks. Mice were then challenged with ultrasonic mesh nebulized LPS 0.05% (wt/vol%) (FLAEM Medical Devices, Flaem^®^ Smarty, MF06E00, Desenzano del Garda, Italy) in vapor chambers at a rate of ≥0.2 mL/min, droplet size <3.3 µm, 20 min per day for 3 consecutive days, followed by once/week for 3 additional consecutive weeks. Mice in the OVA asthma group were treated as above, except 10 μg of OVA (chicken egg, Sigma–Aldrich, Schnelldorf, Germany) was combined with 1% alum adjuvant (Pierce, Rockford, IL, USA) for IP sensitization injections, while 1% (wt/vol%) OVA was used for the challenge phase. The control group was subjected to an identical timeline with sensitization injections containing alum adjuvant while challenges were conducted using 0.9% saline, as illustrated in Appendix A. In summary, these control animals received identical handling and treatments as the asthma test groups, except they were not exposed to antigens ensuring equivalent handling and exposure to stress-related variables.

Mice were humanely killed using carbon dioxide (CO_2_) 24 h after the final allergen challenge, followed by trans-cardiac perfusion with 0.9% saline. The brain and lungs were collected and rapidly frozen using liquid nitrogen prior to storage at −80 °C for further analysis.

### 4.3. Pulmonary Histopathology

Whole lungs were sampled from 6 mice out of the total 7 mice per group, thawed and post-fixed in 10% PFA for 24 h and sent to Monash Histology Platform (Melbourne, Australia) for processing. In brief, post fixation, the lung samples were dehydrated using an ascending series of ethanol: 70% for 24 h, 80% for 30 min, 90% for 75 min, 95% for 75 min and 100% for 60 min. Samples were then cleared via two submersions in xylene for 30 min each and then embedded with paraffin. Paraffin blocks were sectioned at 4 µm thickness on a rotary microtome (Leica Biosystems, Wetzlar, Germany), mounted onto glass microscope slides and stained with hematoxylin–eosin (H&E) as previously described [140]. All the stained lung tissue sections were scanned and then imaged using VS120 Automated Slide Scanner (Olympus, Tokyo, Japan) at the Monash Histology Platform. Signs of lung inflammation and the extent of peribronchial and perivascular inflammatory-cell infiltration were qualitatively scored in a blinded manner [141,142]. Briefly, seven randomly picked lung sections per mouse (9.1 mm long × 5.9 mm wide) were scored using a scale from 0 to 3. A value of 0 was given when no inflammatory cells were detected, a value of 2 when there was a thin layer of inflammatory cells (one to five cell thickness) surrounding most bronchi and blood vessels and a value of 1 where the levels of inflammatory cells was in between these definitions, i.e., when there were a limited number of inflammatory cells detected but were not enough to constitute a thin layer which was one to five cell thickness. Finally, a value of 3 was given when a thick layer of inflammatory cells was detected surrounding most bronchi and blood vessels (more than five cell thickness). Furthermore, Masson’s trichrome stain was used specifically for the detection of sub-epithelial collagen deposition (fibrosis).

### 4.4. Brain Microdissection

Brains (n = 7 from each group) were mounted in Tissue-Plus optimal cutting temperature compound and then sectioned using a cryostat (CM 1860, Leica Biosystems, Germany) as serial 100 μm coronal sections covering the entire rostro-caudal extent of the hypothalamus (bregma 0.14 mm to −2.54 mm) at −20 °C. Each section was mounted on a glass microscope slide, and a circular region of tissue was collected using micro-punch biopsy pens (World Precision Instruments, Hitchin, UK) with different diameters (2 mm and 2.5 mm). The resulting area that was collected corresponded to the majority of the hypothalamus, with minimal inclusion of surrounding regions. To provide a detailed illustration of the collected brain areas, schematic reconstructions of the isolated regions were created based on the Franklin & Paxinos mouse brain atlas [143]. These schematics were then used to refine our predicted collection maps, establishing two boundary regions: the green dotted line represents the areas that were reliably collected in almost all samples, the red dotted line represents the areas that were reliably excluded in almost all samples, while the areas in between these two lines represent a variable collection (Appendix A).

### 4.5. RNA Extraction and Quality Control

Total RNA from the dissected hypothalamus was isolated using the RNeasy Mini Kit (Cat# 74106, Qiagen, Hilden, Germany). Frozen brain tissues were thawed and then homogenized and disrupted in dithiothreitol (DTT) (ThermoFisher Scientific, Waltham, MA, USA, Cat# R0861) mixed with RLT buffer RNeasy Mini kit following the manufacturer’s protocol (10 µL DTT per 500 µL RLT buffer). To enhance RNA binding to the silicon membrane in the spin columns, 70% propanol was used. DNA contamination was reduced using the RNase-Free DNAse kit (Qiagen, Cat# 79254), and the final pure RNA was eluted via the RNase-free water (Qiagen, Cat# 129112). RNA purity was checked using NanoDrop (Thermo Scientific™, USA, Cat# 840274100), while the concentration was measured via the Qubit™ RNA assay kit in Qubit™ 4 Fluorometer (ThermoFisher Scientific, USA, Cat# Q33226). RNA integrity was measured using the Agilent RNA screenTape assay for 4200 TapeStation System (part# G2991BA, Agilent Technologies, Santa Clara, CA, USA), with all samples exhibiting (RIN^e^ > 7.5). RNA concentrations and volumes were normalized (280 ng of RNA in a volume of 28 µL) for all brain samples.

### 4.6. RNA Transcriptome Sequencing

RNA sequencing was conducted at the Deakin Genome Centre (Deakin University, Waurn Ponds campus, Waurn Ponds, VIC, Australia). NEBNext^®^ Ultra™ II Directional RNA Library Prep Kit for Illumina (New England Biolabs, Ipswich, MA, USA, Cat# E7645L) was used for mRNA sequencing library preparation. Briefly, the Illumina specific NEBNext adaptor (15 μM) was diluted fivefold to initiate the adaptor ligation step. The PCR enrichment process involved the repetition of cycling conditions, denaturation and annealing/extension cycle steps for adaptor-ligated DNA, totaling 10 cycles. The libraries were quantified, and their size was estimated using both the Qubit 3.0 Fluorometer and the 4200 TapeStation System (Agilent Technologies, Santa Clara, CA, USA).

In a new microfuge tube, 2 µL from each library was combined and subjected to enzymatic treatment using the Illumina Free Adapter Blocking Reagent (Illumina, San Diego, CA, USA). The MiniSeq Sequencer (Illumina, San Diego, CA) was used to pre-sequence the combined library with 2 × 150 bp paired-end reads to capture the read distribution of individual samples. Subsequently, every library was recombined to achieve uniform molar concentrations and underwent enzymatic treatment, denaturation and normalization to 2 nM. The consolidated library was ultimately subjected to sequencing on the NovaSeq 6000 Sequencer (S2 v.15 kit, 2 × 100 bp paired-end reads) (Illumina, San Diego, CA).

### 4.7. Bioinformatics Analysis

The reference mouse transcriptome was downloaded from GENCODE version 28 [144]. The raw reads (FASTQ) were inspected for sequence quality using Fastqc (v0.11.9) [145] and trimmed using Skewer (v0.2.2) [146] to remove Q < 20 bases from 3′ ends. Kallisto (0.46.1) was used to align the paired-end RNA-seq reads to the mouse transcriptome [147]. The counts at the transcript level were imported into RStudio v4.1 and then consolidated into counts at the gene level. Genes displaying an average count of less than 10 reads across samples were omitted from subsequent analysis. The differential gene expression between control and asthmatic groups was performed using the generalized linear model proposed and implemented in the R package DESeq2 (1.36.0). DESeq2 is one of the most popular statistical tools for differential expression analysis. It makes full use of biological replicate information to estimate differential expression *p*-values and effect sizes, as detailed in the publication by Love et al. [148]. Volcano plots and heatmaps were generated in base R. For pathway analysis, REACTOME gene sets were obtained from the Molecular Signatures Database and converted to mouse gene identifiers with the msigdbr R package (1.4.0) [149,150,151]. Differential pathway analysis was then performed with the mitch R package (1.8.0) with default settings [152]. Mitch is a functional class scoring technique that uses a rank–ANOVA test to ascertain the collective enrichment of genes in either the up- or downregulated direction. To reduce the chance of false positives, differential gene and pathway analyses were subjected to false discovery rate (FDR) correction using the method of [153]. Genes and pathways with an FDR < 0.05 were considered statistically significant. All the data were analyzed using RStudio version (2022.07.1) and GraphPad prism version 8 software.

### 4.8. Literature Search

Manuscripts that potentially demonstrated changes in expression in our gene sets in experimental groups relevant to mental health or neuroinflammation were initially examined by searching for keywords within relevant databases.

Our literature research strategy focused on the articles that investigate gene expression changes in the context of both neuroinflammation and mental health. We combined the keywords “neuroinflammation” and “mental health” with each of the dysregulated gene sets (e.g., neuroinflammation AND post-chaperonin-tubulin folding pathway) OR (e.g., mental health AND post-chaperonin-tubulin folding pathway). The PubMed, Scopus and ScienceDirect databases were used to extract the eligible studies. We also identified publications by searching the study design: either observational study “gene expression studies” (the differences in the expression of the gene set between two groups test and control) and pharmacogenetic study (alteration of the gene set in response to drug that is involved in mental health or neuroinflammation). Moreover, we included studies that investigated preclinical in vivo and in vitro models as well as human clinical trials. Keywords included the following: neuroinflammation, or mental health, or neuropsychiatric disorders, or depression, or anxiety. Our research was limited to English highly cited and peer-reviewed articles. After the above approach identified potential hits, each manuscript was individually assessed. The manuscripts cited in Appendix A and in the discussion section were confirmed to present data in which the specific gene set exhibited altered expression with asthma-challenged groups relevant to mental health or neuroinflammation compared to controls.

Finally, a comprehensive diagram was drawn to include the top altered significant gene sets, their upstream functional hierarchy and their downstream consequences. The overarching goal was to unravel the complex interplay between the altered gene sets and mental health/neuroinflammation outcomes.

## 5. Conclusions

Our hypothalamic transcriptomic findings demonstrated the possible molecular mechanisms within the hypothalamus underlying the lung–brain axis in response to lung inflammation/asthma. Importantly, changes in gene sets relevant for neuroinflammation and mental health were identified. The identification of specific genetic signatures not only assists in our understanding of the pathophysiological aspects of asthma-related mental health disorders but also creates new opportunities for the exploration of potential therapeutic targets that may alleviate the complex interconnection between asthma and mental health.

## Figures and Tables

**Figure 1 ijms-25-07391-f001:**
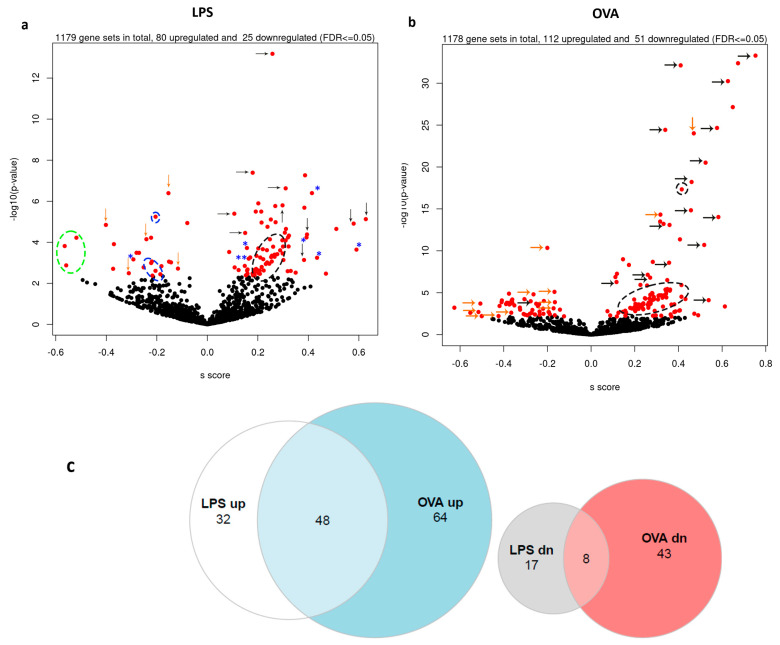
Effect of LPS− and OVA−treated groups on mouse transcriptome gene set analysis in the hypothalamus. (**a**,**b**) Volcano plot of gene set enrichment analysis for the LPS− and OVA−treated groups, respectively. The black and orange arrows indicate the altered gene sets that are associated with metabolism and nervous system function, respectively. Black annotated circles refer to inflammatory and cytokine gene sets, blue circles indicate immune system gene sets, and the green circle refers to the homeostasis gene sets. Autophagy and cellular response to stimuli gene sets are indicated by the blue asterisks. Please refer to Appendix A for detailed information. (**c**) Venn diagram showing the common gene pathways with differential expression.

**Figure 2 ijms-25-07391-f002:**
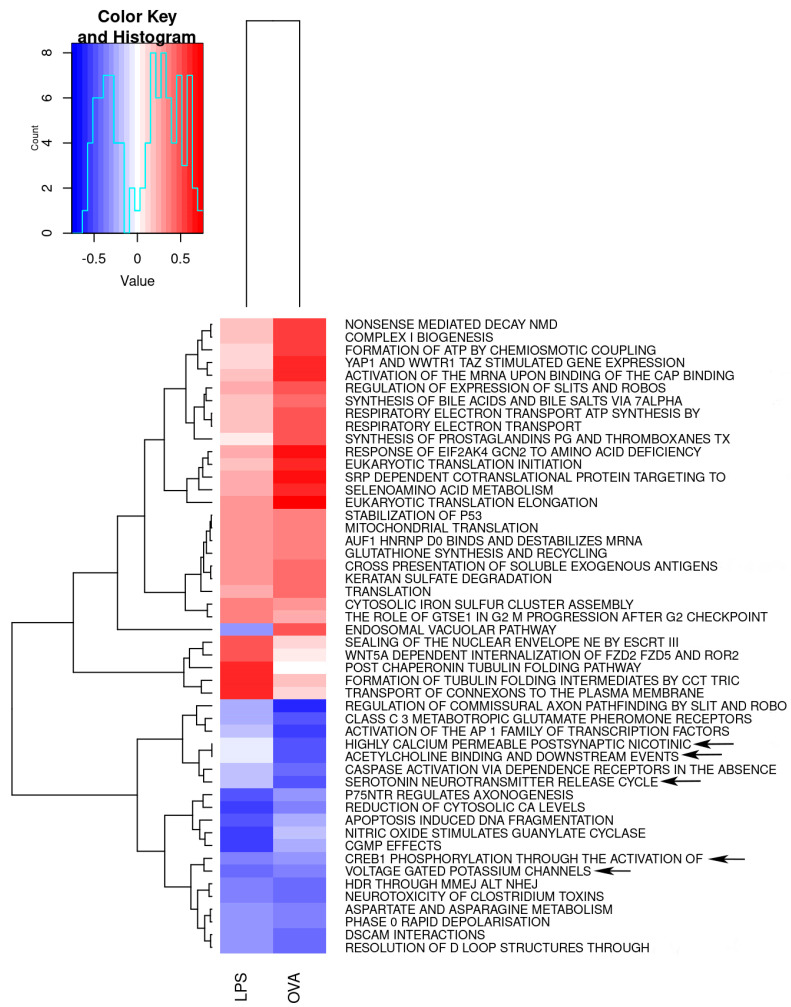
Heatmap for top 50 joint dysregulated gene sets by magnitude of change in asthma relative to the control group. Upregulated and downregulated gene sets are indicated in red and blue, respectively. Among the top 50 dysregulated gene sets, 21 were categorized under metabolism. Of these, 8 are related to protein metabolism, including pathways for translation and protein folding. Four gene sets are directly associated with the aerobic respiration process, while the remaining sets are linked to the metabolism of carbohydrates, RNA and lipids, highlighting the brain’s metabolic response to asthma. The remaining gene sets are directly linked to signal transduction (3 gene sets), programmed cell death (2 gene sets), cell cycle (3 gene sets), hemostasis (3 gene sets), gene expression (1 gene set) and cellular response to stimuli (1 gene set). Of interest, 5 gene sets were found to be directly involved in neuronal systems (black arrows).

**Figure 3 ijms-25-07391-f003:**
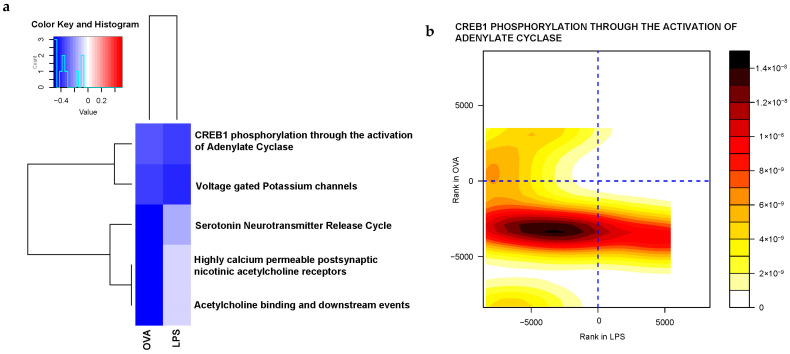
Integrated analysis identifies altered neuronal signaling gene sets in the hypothalamus in response to asthma challenge. (**a**) Heatmap for the five significant downregulated (blue) neuronal gene sets common to both LPS− and OVA−challenged groups. (**b**–**f**) Contour plots of these gene sets, where the X-axis represents gene rankings based on their fold change in the LPS group, and the Y-axis represents the rankings of the same genes based on their fold change in the OVA group, showing consistent downregulation of member genes in both asthma groups (dark red in all lower left quadrants). (**b**) *CREB1 phosphorylation through the activation of adenylate cyclase*, (**c**) *voltage gated potassium channels*, (**d**) *serotonin neurotransmitter release cycle*, (**e**) *highly calcium permeable postsynaptic nicotinic acetylcholine receptors*, (**f**) *acetylcholine binding and downstream events*.

**Figure 4 ijms-25-07391-f004:**
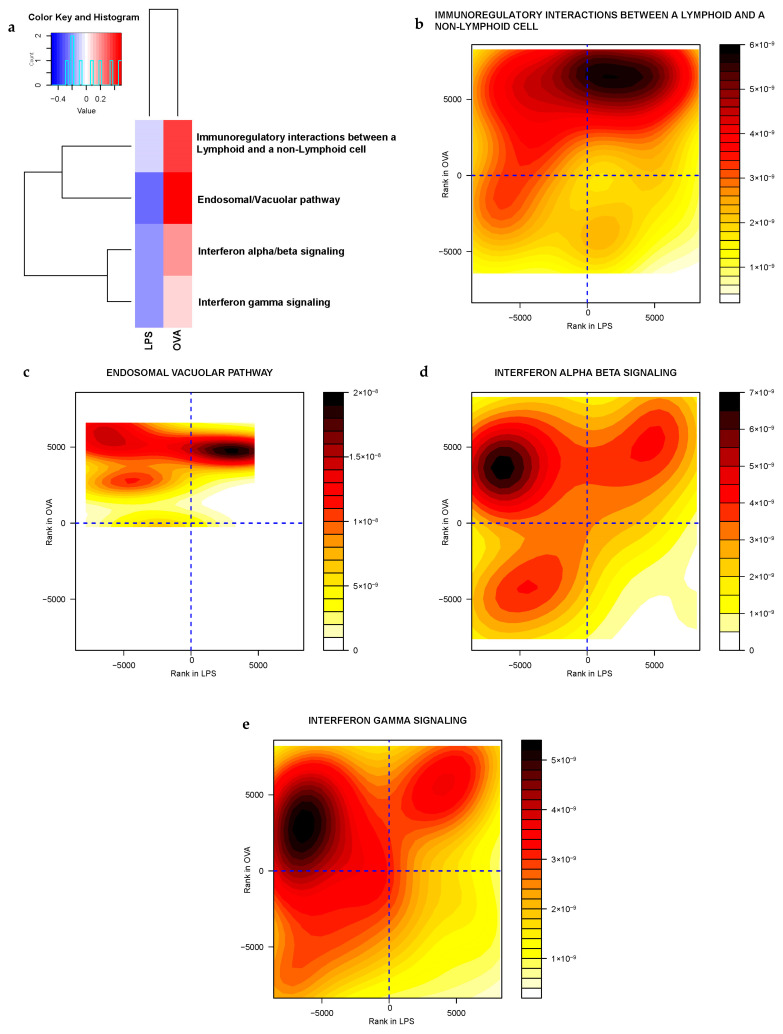
Integrated discordant analysis identifies differences in immune responses between LPS− and OVA−challenged groups in the hypothalamus. (**a**) Heatmap of the four adaptive immune signaling gene sets significantly downregulated (blue) in the LPS−challenged group and upregulated (red) in the OVA−challenged group. (**b**–**e**) Contour plots of these gene sets, where the X-axis represents gene rankings based on their fold change in the LPS group and the Y-axis represents the rankings of the same genes based on their fold change in the OVA group. A positive rank on the Y-axis coupled with a negative rank on the X-axis suggests an inverse relationship in the response of the immune system genes to LPS and OVA challenges. (**b**) *Immunoregulatory interactions between a lymphoid and a non-lymphoid cell*, (**c**) *endosomal/vacuolar*, (**d**) *interferon alpha/beta signaling*, (**e**) *interferon gamma signaling*.

**Figure 5 ijms-25-07391-f005:**
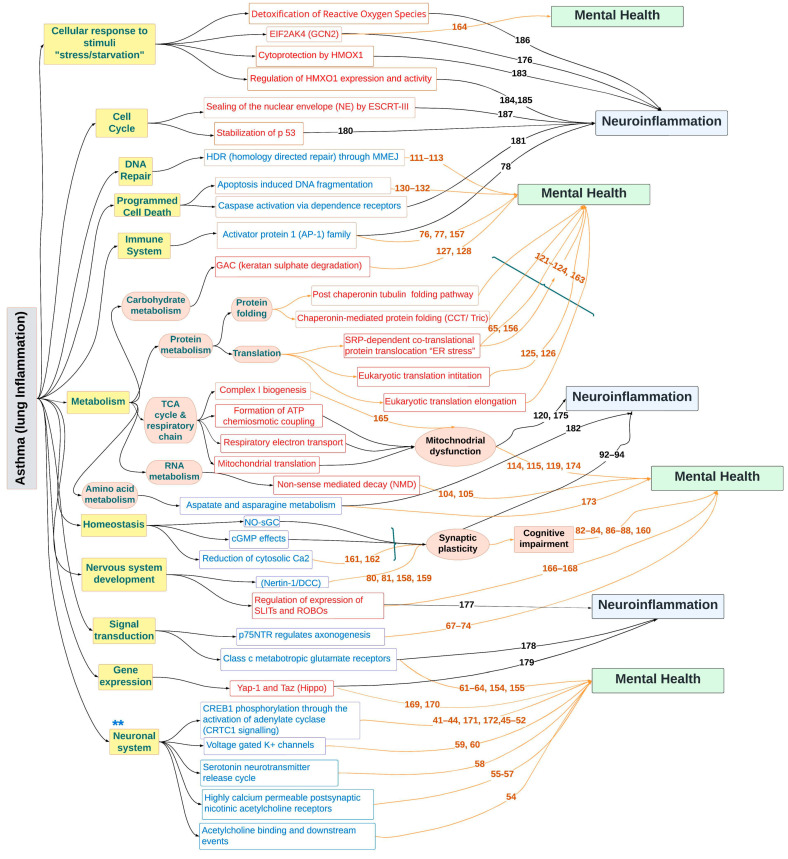
The prospective molecular basis of the lung−brain crosstalk within the hypothalamus underlying neuroinflammation and mental health problems. The top selected gene sets that were altered in the hypothalamus in response to asthma induction are depicted. Key elements of the upstream functional hierarchy are highlighted in yellow on the left. In the middle, gene sets that are upregulated are shown in red, and those that are downregulated are shown in blue. The right side depicts downstream effects associated with neuroinflammation (blue boxes) and mental health (green boxes). Of particular interest are five gene sets associated with neuronal signaling (**). Previous studies linking these gene sets to mental health or neuroinflammation are indicated by orange and black numbers, respectively. Detailed reference information is available in Appendix A.

## Data Availability

The data that support the findings of this study are available from the corresponding author on reasonable request. Scripts used in this analysis are available at the GitHub repository (https://github.com/markziemann/lung_brain) accessed on 23 February 2022.

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
