# Peer review of "Novel Insights into Changes in Gene Expression within the Hypothalamus in Two Asthma Mouse Models: A Transcriptomic Lung–Brain Axis Study"

_ijms, 2024, doi:10.3390/ijms25137391_

Round 1

Reviewer 1 Report

Comments and Suggestions for Authors

After reviewing the manuscript titled "Novel Insights into Changes in Gene Expression Within the Hypothalamus in Two Asthma Mouse Models: A Transcriptomic Lung-Brain Axis Study," I suggest some improvements regarding various aspects, including experimental design, data analysis and interpretation of results:

1.     Clearly state the specific hypotheses and objectives at the beginning of the introduction. A clear hypothesis helps guide the reader and establishes the context for the study, ensuring that the objectives are well-defined and measurable.

2.     Provide a more detailed justification for the choice of BALB/c mice and the relevance of this model to human asthma and mental health conditions. Justifying the animal model strengthens the rationale behind the study and ensures that the findings are more relatable to human conditions.

3.     Elaborate on the specific parameters used for LPS and OVA administration, including dosages, frequency, and duration of exposure. Detailed methodology allows for reproducibility and helps other researchers replicate the study or compare their methods with those used in this study.

4.     Provide more information about the control group, including any handling or treatment they received to account for potential stress-related variables. A well-defined control group is essential for validating the experimental results and ensuring that observed effects are due to the treatments rather than other factors.

5.     Include a justification for the statistical methods used, particularly for differential gene expression analysis and enrichment analysis. Justifying the choice of statistical methods ensures that the analyses are appropriate for the data.

6.     Clearly state how biological replicates were integrated into the analysis and how variability was addressed.

7.     Provide a more detailed explanation of the REACTOME pathway analysis, including how gene sets were selected and interpreted. Detailed pathway analysis helps readers understand the findings' biological significance and implications for lung-brain crosstalk.

8.     Offer a more thorough interpretation of the heatmap and volcano plot results, highlighting key genes and pathways relevant to asthma and mental health. Interpretation of these visual tools aids in comprehending complex data and identifying the most significant findings.

9.     Compare and contrast the findings from single gene analysis and gene set analysis, discussing why certain approaches might yield different insights. Understanding the strengths and limitations of different analytical approaches provides a more comprehensive view of the data.

10.  Include a section on validating key findings using independent methods like qPCR or Western blotting. Validation with independent methods increases confidence in the RNA-seq results and supports the study's conclusions.

11.  Discuss the potential impact of sex differences on the study findings and consider including both male and female mice in future studies. Considering sex differences ensures that the findings are more broadly applicable and address a significant variable in biological research.

12.  Expand the discussion on how specific neuroinflammatory pathways identified in the study might contribute to mental health disorders in asthma patients. Linking specific pathways to mental health outcomes provide deeper insights into the mechanisms of lung-brain crosstalk and potential therapeutic targets.

13.  Where possible, relate the mouse model findings to human studies and discuss the translational relevance of the results. Connecting the findings to human studies enhances their significance and potential impact on human health.

14.  Provide a more comprehensive discussion of the study's limitations, including potential confounding factors and areas for future research.

Author Response

Many thanks.

Reviewer 2 Report

Comments and Suggestions for Authors

The study by Bastawy et al. aimed to examine changes in gene expression in the hypothalamus following lung inflammation (asthma) in BALB/c mice using two asthma mouse models (LPS and OVA). Bastawy et al. demonstrate that LPS exposure and OVA exposure both result in lung inflammation. They also demonstrate that while both LPS exposure and OVA exposure result in changes in similar gene sets (e.g., signal transduction metabolism, immune response, neuroplasticity, etc.), there were also unique gene sets only altered in one of the two asthma models, with LPS exposure being the only asthma model to show statistically significant single gene changes. Taken together, the study by Bastawy et al. is interesting, but there are controls that are missing that make the conclusions drawn questionable. 

Major concerns:

1) How do the authors know that "asthma" or "lung inflammation" is causing the gene expression changes in the hypothalamus? Without the proper controls (i.e., gene expression analysis of "sensitization (IP injection)" alone vs. "challenge (vapour chamber)" alone vs. "sensitization + challenge"--needs to be compared), the authors can't say that the IP injection alone (or vapour chamber alone) aren't causing the changes in gene expression observed. Especially as IP injection of LPS is known to have impacts on gene expression in the brain when administered by itself (many different infection models perform LPS IP injections in mice); therefore, the authors are unable to make conclusions such as "This study serves as a new reference for the transcriptomic changes in the hypothalamus in response to lung inflammation (asthma)" until all the proper controls are performed/analyzed.

2. Given that asthma is more common in females, why did the authors only select to perform these experiments in male mice? There is no mention anywhere in the manuscript why this decision was made or sex differences in asthma rates in each sex or that only using males is a limitation of this study. The authors should have performed this analysis in both male and female mice, and at the very least, explain their reasoning for not doing so.

Minor concerns:

1. The authors do not validate any of the findings for their RNAseq datasets. It would be very helpful for the authors to use fluorescence in situ hybridization technology (e.g., RNAScope) to examine prominent gene changes (e.g., gene candidates associated with mental health issues) observed (e.g., single genes from LPS exposure if needed) in the hypothalamus to determine where/what hypothalamic nuclei these changes are occurring in (e.g., PVN) and what cell types (e.g., CRH neurons) might be showing these changes (e.g., inform on important signalling changes to stress pathway, etc.). 

2. The study would be strengthened if the authors threw on IBA1 (microglia) and GFAP (astrocytes) to get a basic view of their numbers/morphology in the hypothalamus of "sensitization" mice alone vs "challenged" mice alone vs "sensitization + challenged" mice.

Author Response

Many thanks,

Round 2

Reviewer 1 Report

Comments and Suggestions for Authors

The author's answers to my previous comments and the improved version of the manuscript determine my recommendation of acceptance.

Reviewer 2 Report

Comments and Suggestions for Authors

The revised manuscript submitted by Bastawy et al. did not address the concerns that were raised by the Reviewers, and instead Bastawy et al. only only slightly modified the text and added some additional text to the discussion.

In order to address the Reviewer's concerns, Bastawy et al., needs to conduct additional experiments to validate their findings, and when doing so, could easily look to see if they apply to both sexes (e.g., qPCR, Western blot, IHC and/or RNAScope, etc., in both males and female) and conduct the proper controls (e.g., OVA alone, LPS alone, etc.). 
